# The Expression of miRNAs in Human Ovaries, Oocytes, Extracellular Vesicles, and Early Embryos: A Systematic Review

**DOI:** 10.3390/cells8121564

**Published:** 2019-12-04

**Authors:** Albert Salas-Huetos, Emma R. James, Kenneth I. Aston, Timothy G. Jenkins, Douglas T. Carrell, Marc Yeste

**Affiliations:** 1Andrology and IVF Laboratory, Division of Urology, Department of Surgery, University of Utah School of Medicine, Salt Lake City, UT 84108, USA; Emma.James@utah.edu (E.R.J.); ki.aston@hsc.utah.edu (K.I.A.); Tim.Jenkins@hsc.utah.edu (T.G.J.); douglas.carrell@hsc.utah.edu (D.T.C.); 2Department of Human Genetics, University of Utah School of Medicine, Salt Lake City, UT 84108, USA; 3Department of Obstetrics and Gynecology, University of Utah School of Medicine, Salt Lake City, UT 84108, USA; 4Biotechnology of Animal and Human Reproduction (TechnoSperm), Unit of Cell Biology, Department of Biology, Institute of Food and Agricultural Technology, Faculty of Sciences, University of Girona, 17003 Girona, Spain

**Keywords:** miRNAs, extracellular vesicles, ovary, oocyte, embryo

## Abstract

The recent discovery of microRNAs (miRNAs) in human reproductive tissues and cells indicates a possible functional role in reproductive function. However, the studies published to date in female reproductive tissues/cells and embryos are inconclusive and sometimes controversial. In order to update the knowledge of this field, the present study aimed to discuss, through a systematic review, the role of miRNAs in female human reproduction and early embryogenesis. We conducted a systematic review of the published literature in MEDLINE and EMBASE databases through June 2018 (plus a complementary search until July 2019), in accordance with the PRISMA guidelines. We have included descriptive and observational studies, in which fertile/infertile women were well-defined. The primary outcome was the miRNA expression in ovaries, oocytes, extracellular vesicles, and embryos. We identified 25,204 articles, of which 28 were selected for qualitative analysis: 18 in ovaries and extracellular vesicles, three in oocytes, and seven in embryos. The present systematic review of descriptive and observational studies demonstrates that aberrant miRNA expression in female reproductive tissues/cells and embryos is related with infertility and embryogenesis errors. The expression of specific miRNAs, particularly in extracellular vesicles, may be used in the future as biomarkers of infertility and prognostic tools of embryo development.

## 1. Introduction

MicroRNAs (miRNAs) are single-stranded RNA molecules from 16–28 nucleotides in length (www.mirbase.org) that function to regulate gene expression by means of the formation of semi-complementary structures between mRNA and miRNAs in 3′ untranslated regions (3′UTR) (in plants the mRNA-miRNA interaction is near perfect, whereas in animals, miRNAs bind in semi-complementarity with mRNA encoding regions) [1,2]. Each miRNA has hundreds of potential mRNA targets, due to the semi-complementary configuration [3]. Mediated by Argonaute (AGO) proteins and components of the RNA-induced silencing complex (RISC), miRNAs can act by the following mechanisms: (1) deadenylation of poly-A tails or, (2) interaction/inhibition with/of translation initiation factors [4,5].

A very recent in silico high-throughput experimental strategy estimated that about 2,300 mature miRNAs exist in humans, 1,982 of which are currently annotated in miRBase v.22.1 (www.mirbase.org). The most recent version of miRBase (v22.1) contains 2,654 mature human miRNA sequences [6]. In 2002, a landmark article reported the presence of small RNAs in sperm cells [7]; later, the same authors described that they were probably miRNAs [8]. In parallel, other authors described the presence of miRNAs in all human reproductive tissues and cells [9,10], evidencing the role in male [11] and female [12] gametogenesis and embryo development [13,14]. The presence of miRNAs in these cells and tissues, as well as the known functions associated with these molecules, suggests that a deregulation of their expression may result in alterations in reproductive system and potentially in early embryogenesis.

Therefore, we conducted the first systematic review of high-quality studies published to date to clarify the existing association between miRNA expression in different female reproductive tissues and cells as well as the early embryo, and the potential associations with infertility and early embryogenesis. Although a meta-analytic analysis is not possible due to the heterogeneity of published articles, a well-designed systematic review is beneficial in consolidating the current body of evidence, which is sometimes inconclusive and controversial. This review is aimed at helping researchers to identify deficiencies and test new hypotheses on this field.

## 2. Material and Methods

### 2.1. Search Strategy and Systematic Review Registration

A systematic search was performed in the MEDLINE-Pubmed (http://www.ncbi.nlm.nih.gov/pubmed) and the EMBASE (https://www.embase.com/#search) databases. The search was performed on several occasions with the last screening search taking place on June 2018 (and a complimentary search until July 2019) without further restrictions on publication date in accordance with the guidelines and checklist of the Preferred Reporting Items for Systematic Reviews and Meta-Analyses (PRISMA) [15]. The protocol has been registered in the PROSPERO registry (PROSPERO 2018: CRD42018099793; http://www.crd.york.ac.uk/PROSPERO).

The search query and screening strategy aimed at collecting all records related to human infertility, and miRNAs using both Medical Subject Headings (MeSH) and keywords. The complete search strategy and filters included is available in Appendix A. 

### 2.2. Study Selection and Eligibility

Titles and abstracts were screened in duplicate by two of the six authors (A.S-H. and M.Y.) for eligibility. A third author resolved disagreements (E.R.J.). A PICOS (Patient, Intervention, Comparator, Outcome, Study) design structure was used to develop the study questions and the inclusion/exclusion criteria (Appendix A). Studies were eligible for inclusion if they were human studies (descriptive, case-control, cross-sectional, observational prospective, and retrospective studies) in which fertile/infertile conditions were well-defined. The primary outcome was the miRNA expression level. Comparators were always a well-defined control/fertile group without interventions beyond the intrinsic IVF/ICSI procedures in some cases. After primary screening (evaluation of the scope of the study based on the title and abstract), the full texts of the selected articles were obtained. Regarding eligibility, randomized clinical trials (RCTs), animal studies, review articles, studies evaluating free circulating miRNAs, studies with gestational or specific female sexual tract tissues (e.g., fallopian tubes or endometrial tissue), studies in male reproductive cells/tissues, and studies with other types of small RNAs not included in the miRNA category were excluded. We conducted a complementary search of reference lists of articles and key journals for human studies after June 2018 until July 2019. After the secondary screening, and once compliance with all the inclusion/exclusion criteria was assessed, more full texts were obtained.

### 2.3. Data Extraction and Quality Assessment

We extracted the following information from each study: authors, year of publication, journal, title of the article, localization of the study, study participant age, infertility phenotype, number of participants (sample size), study design, primary outcome and aim of the study, and major findings. The NHLBI-NIH quality assessment tool was used to assess the quality of each study (https://www.nhlbi.nih.gov/health-topics/study-quality-assessment-tools). Similar to the publication selection process, disagreements were resolved by consensus between three authors (A.S-H., M.Y., and E.R.J.). The scale assesses different domains depending on study type. The complete quality assessment criteria are available in Appendix A. Studies with a score ≥3 points for descriptive studies, ≥4 points for case-control studies and ≥5 points for retrospective/prospective and cross-sectional studies were considered of modest to good quality and were eligible for the present systematic review. 

## 3. Results

### 3.1. Identification and Selection of the Articles

We identified 16,469 articles after a primary search by MEDLINE-Pubmed, and 8,735 articles after a primary search by EMBASE database (Figure 1). After analyzing the results (*n* = 25,204), we immediately excluded 2,569 records because they were duplicated (*n* = 819); they were conference papers (*n* = 1,559), comments, replies or letters to the Editor (*n* = 9); they were retracted papers (*n* = 2), or because they were not written in English (*n* = 180). The remaining 22,635 records were evaluated based on their title and abstract, and 22,343 were excluded. The remaining 292 articles were collected as full-texts, and 32 additional manuscripts were included after a complementary search. Therefore, the inclusion/exclusion criteria and quality scores were assessed in 324 full-text articles. One hundred and seventy-six of these 324 articles were excluded because they did not meet the inclusion/exclusion criteria, while 70 articles were excluded because they were review articles, two articles were excluded because their full-text was not available, 41 articles were excluded because they were studies in male reproductive tissues/cells, and seven articles were excluded because they did not reach the quality assessment threshold. Finally, after applying all of the eligibility parameters, 28 articles were used for qualitative analysis.

### 3.2. Summary of Selected Studies

Of the 28 articles included, 18 evaluated the miRNAs in ovaries and/or extracellular vesicles (i.e., exosomes and microvesicles) [9,16,17,18,19,20,21,22,23,24,25,26,27,28,29,30,31,32]; three in oocytes [33,34,35]; and seven in embryos [36,37,38,39,40,41,42].

#### 3.2.1. Ovaries and/or Extracellular Vesicles (Exosomes and Microvesicles)

Eighteen studies investigating miRNAs found in ovaries and extracellular vesicles isolated from follicular fluid are included in this systematic review and are summarized in Table 1A. The quality scores of these articles are the following: descriptive studies mean = 5/9, case-control studies mean = 5.9/12, prospective studies mean = 7.5/14, and cross-sectional studies mean = 8/14.

Four of the 18 studies were conducted in fertile women and aimed at characterizing the normal miRNA expression patterns in human ovaries and exosomes (descriptive studies). Liang and collaborators, using qRT-PCR described a total of 216 miRNAs expressed in a healthy human ovary, with miR-26a being the most highly expressed in ovaries [9]. Although this was the first descriptive study, published results must be cautiously interpreted as they represent only a single ovary. Another study involving 20 women undergoing ICSI (some with PCOS) found 538 miRNAs in human follicular fluid (including miR-19b, miR-24, and miR-222). With a case-control approach analysis, the same study also discovered that the expression levels of miR-132 and miR-320 were significantly lower in controls (*n* = 22) than in PCOS (*n* = 22) patients [22]. Another study that evaluated follicular fluid exosomes isolated from 15 women undergoing ICSI found, using qRT-PCR, that 37 miRNAs were more highly expressed in human follicular fluid than in blood plasma and also identified 22 miRNAs that were exclusively present in the exosomes [23]. Finally, Tong and collaborators investigated, through NGS, the miRNA expression profiles of corona radiata (CRCs) and cumulus oophorus cells (COCs) in five women and identified a total of 785 and 799 miRNAs in CRCs and COCs, respectively. In addition, 72 DE-miRNAs were identified when comparing CRCs with COCs [24].

We identified two prospective studies aimed at comparing, through qRT-PCR, the expression of a set of ovarian miRNAs between older and younger women. The first work, which involved 16 women (young age: < 31 years (*n* = 8) and advanced age: >38 years (*n* = 8)), reported that miR-21-5p is present at significantly higher levels in the follicular fluid of younger women, and miR-99b-3p, miR-134 and miR-190b are found at significantly higher levels in the follicular fluid of older women [26]. The second work investigating follicular fluid miRNAs based on oocyte stage followed a more complex methodology that involved a two-stage analysis: 1) younger group: <35 (*n* = 15) vs. older group: >37 (*n* = 15), and 2) metaphase II (MII) vs. germinal vesicle (GV) (*n* = 12), and metaphase I (MI) vs. MII (*n* = 9); each patient acted as her own control. In this case, miR-424 was found to be highly expressed in the follicular fluid of patients with advanced age. Moreover, 13 DE-miRNAs were identified in the follicular fluid of MII- vs. GV-oocytes, and seven DE-miRNAs were detected when MI- and MII-oocytes were compared [32].

Only one cross-sectional study could be included in this review. In that study, using qRT-PCR, the authors investigated miRNA content of follicular fluid microvesicles isolated from follicles corresponding to oocytes with normal fertilization (*n* = 93) compared with those corresponding to oocytes that failed to fertilize (*n* = 33). These authors found 12 DE-miRNAs between normal and failed-to-fertilize groups. MiR-92a and miR-130b were over-expressed in follicular fluid microvesicles associated with oocytes that failed to fertilize compared to those that were normally fertilized. Interestingly, miR-888 was over-expressed and miR-214 and miR-454 were under-expressed in samples that resulted in low quality compared to top-quality day-3 embryos [31].

Eleven case-control studies were included in this systematic review. Of those that were included, two different groups of articles were identified: 1) studies focused on the miRNAs expressed in patients with PCOS; and 2) studies with poor responder patients, diminished ovarian reserve (DOR) or no blastocyst formation.

Eight studies focused on the miRNAs expressed in patients with polycystic ovary syndrome (PCOS) were identified. Four out of the eight works were designed as high-throughput studies (two used NGS plus validation with qRT-PCR and two used qRT-PCR only). Luo et al., 2019 [30] published a very recent article comparing the miRNA expressed in granulosa cells (GC) of poor ovarian response (*n* = 7) and of PCOS patients (*n* = 20) with those found in control individuals (*n* = 18). They identified 20 conserved and three novel miRNAs that were upregulated in the poor ovarian response group, and 30 conserved miRNAs and one novel miRNA that were up-regulated in the PCOS group [30]. In a similar GC-based study, Xu et al., 2015 [21] compared, through NGS, 21 women presenting PCOS with 20 women without PCOS (control). These authors identified 59 known DE-miRNAs in PCOS patients; 21 miRNAs were up-regulated and 38 miRNAs were down-regulated in PCOS patients [21]. In another work, Roth et al., 2014 [17] compared, through qRT-PCR, the miRNAs from follicular fluid of 10 PCOS patients with that of 10 oocyte donors and discovered 29 DE-miRNAs in PCOS patients. In the validation step, only the levels of five of ten validated miRNAs (miR-9, miR-18b, miR-32, miR-34c, and miR-135a) were significantly higher in PCOS patients compared with oocyte donors [17]. Finally, the fourth study evaluated the miRNAs expressed in COCs from 24 PCOS patients and 24 controls, and found that the levels of miR-483-5p and miR-486-5p were significantly lower in PCOS than in non-PCOS patients [19].

The other four articles explored a small number of miRNAs in PCOS patients, all using qRT-PCR. One of these studies compared the expression of five miRNAs in the follicular fluid between 30 women with PCOS and 91 women with normal ovarian reserve (NOR; as controls), and found that miR-30a was significantly up-regulated and miR-140 and let-7b were significantly down-regulated in the follicular fluid from patients with PCOS compared with women with NOR [18]. Another work investigated, by comparing women suffering from PCOS (*n* = 25) with women without PCOS (*n* = 20), the effects of miR-145 on cell proliferation and its underlying mechanism in isolated human granulosa cells from aspirated follicular fluid. In this study, miR-145 was found to be down-regulated in PCOS compared to control subjects [16]. In the third study, Eisenberg et al., 2017 [27] evaluated the expression of miR-200b and miR-429 in granulosa cells of 15 normally ovulating women (fertile women) suffering infertility due purely to male infertility factors (infertile men), 18 women with PCOS and seven normally ovulating women. The authors discovered very low expression of both miRNAs in granulosa cells of all patients, with miR-200b expression being relatively higher than miR-429 [27]. In the fourth study, Wang and collaborators measured the expression of miRNAs in the granulosa cells of PCOS patients (*n* = 17) and 17 healthy controls. They identified 21 miRNAs up-regulated and 38 down-regulated in the granulosa cells of PCOS compared to those of healthy patients. Remarkably, the expression of miR-27a-3p was found to be significantly increased in both granulosa cells and ovaries of patients with PCOS compared to healthy controls [20].

In studies with poor responder patients, diminished ovarian reserve (DOR), or no blastocyst formation, three articles were selected. In these three studies, different techniques were used (NGS, microarrays, and qRT-PCR). In the first selected article, Chen et al., 2017 [25] used NGS to compare miRNA expression in the COCs of 10 women with DOR and 10 women with NOR. Seventy-nine annotated miRNAs and five novel miRNAs were found to be differentially expressed between DOR and NOR [25]. In the second work, Karakaya et al., 2015 [29] compared the miRNA expression in COCs of 24 poor ovarian hyperstimulation responders with that of 32 normal responders (controls). They identified 16 miRNAs up-regulated and 88 down-regulated in poor responders (microarray analysis), and validated by qRT-PCR the increased expression of miR-21-5p and the reduced expression of miR-21-3p in poor responders. [29]. Finally, Fu et al., 2018 [28] used qRT-PCR to compare the expression of miRNAs in the follicular fluid in 53 cases that yielded no blastocysts on day-five and 38 controls with viable blastocysts (higher than 3BC/3CB). They observed 13 up-regulated miRNAs and three down-regulated miRNAs in cases (i.e., no blastocyst development) compared with controls. It is worth noting that miR-663b expression was found to be significantly higher in cases than in controls in the validation [28].

#### 3.2.2. Oocytes

Only three retrospective oocyte studies using microarray screening and qRT-PCR validation were included in this systematic review (Table 1-B). The quality scores of these retrospective studies are 7/14 (mean).

With the objective to identify DE-miRNAs and expression patterns of specific miRNAs during meiosis in human oocytes, Xu and collaborators [34] used 392 GV-oocytes and 43 MII-oocytes in a study that involved 251 ICSI cycles. The authors observed that, compared to GV-oocytes, the MII counterparts exhibited an up-regulation of four miRNAs (miR-193a-5p, miR-297, miR-625, and miR-602), and a down-regulation of 11 miRNAs (miR-888*, miR-212, miR-662, miR-299-5p, miR-339-5p, miR-20a, miR-486-5p, miR-141*, miR-768-5p, miR-376a, and miR-15a) [34]. The other work used 36 in vivo matured MII-oocytes from 30 healthy women recruited for oocyte donation and evaluated whether miRNA expression varies with women’s age and ovarian reserve. The study revealed a set of five miRNAs (ENSG00000221162, miR-220b, ENSG00000239174, miR-4262, and miR-1260a) that were more highly expressed in the old than in the young age group, regardless of the ovarian reserve [33]. Finally, Battaglia et al., with the aim to identify whether aging can alter the expression of miRNAs in human oocytes, found that 12 miRNAs were differentially expressed in women of advanced reproductive age (e.g., let-7b, let-7e, miR-19a, miR-29a, miR-126, miR-136, miR-192, miR-203, miR-371a-3p, miR-484, miR-494, and miR-519d) [35].

#### 3.2.3. Embryos

Table 2 summarizes the embryo studies included in this systematic review. The quality scores are the following: case-control studies mean = 5.8/12, and prospective studies mean = 7/14.

Only two prospective studies were included in this systematic review. These works aimed at determining whether human blastocysts secrete miRNAs into culture media and whether these secreted miRNAs reflect the embryonic ploidy status and are predictive of IVF outcomes. First, evaluating miRNA isolated from spent culture media from 28 tested blastocysts (SBM), Rosenbluth and collaborators identified, through qRT-PCR, 10 miRNAs (miR-106b, miR-191, miR-30c, miR-372, miR-376a, miR-548a-3p, miR-548c-3p, miR-548d-3p, miR-576d-3p, and miR-603) consistently expressed in the culture media (miR-372 and miR-191 confirmed by single assay). These authors also found that levels of miR-191 were higher in media from aneuploid embryos, and miR-191, miR-372, and miR-645 were more highly concentrated in day-five media from embryos corresponding to failed IVF cycles [42]. The second study used trophectoderm (TE) samples and their corresponding SBM from five good-quality human blastocysts with qRT-PCR screening and subsequent validation [41]. In that case, the comparative analysis of TE and SBM samples revealed that 96.6% (57 of 59) of the miRNAs detected in the SBM were expressed in TE cells. miRNA analysis of SBM from euploid implanted and non-implanted blastocysts found that expression of two miRNAs (miR-20a and miR-30c) was higher in the former than in the latter [41].

Five case-control studies were included in this systematic review. The first two articles used embryonic tissues (mostly trophoblast) from patients suffering from tubal ectopic pregnancy (EP) and voluntary termination of pregnancy (VTOP; controlled abortion). One of these studies was designed as a screening study (microarray and qRT-PCR validation) with 23 EP cases and 29 VTOP controls and found four miRNAs (miR-196b, miR-30a, miR-873, and miR-337-3p) were down-regulated in EP vs. control samples, and three miRNAs (miR-1288, miR-451, and miR-223) were up-regulated in EP compared to control samples (validation confirmed the DE-miRNAs miR-196 and miR-223) [36]. The second study explored LIN28/let-7 system in 17 EP patients and 23 VTOP controls. They observed that *LIN28b* mRNA was barely detectable in embryonic tissue from early stages of gestation and increased thereafter at seven to nine weeks of gestation. In contrast, expression levels of let-7, miR-132, and miR-145 were high in embryonic tissues from early gestations (≤6-weeks) and abruptly declined thereafter, especially for let-7 (opposite trends for miR-323-3p). Embryonic expression of *LIN28b* mRNA was higher at early stages (≤6-weeks) of EP than in normal gestation. In contrast, let-7a expression was significantly lower in early EP, whereas miR-132 and miR-145 levels were not altered (miR-323-3p expression was also suppressed in ectopic embryonic tissue) [40].

The first article that was published with the aim to examine whether the miRNA expression profile in human blastocysts is correlated with infertility was produced by McCallie et al. (2010) [37]. Evaluating cryopreserved blastocysts with male factor infertility alone (*n* = 6), blastocysts from women with PCOS (*n* = 6), and blastocysts from donor oocyte cycles with no known male factory infertility (*n* = 10), these authors found, through qRT-PCR, that blastocysts derived from PCOS and from infertile males exhibiting significantly lower expression of six miRNAs (let-7a, miR-19a, miR-19b, miR-24, miR-93, and miR-94) than fertile controls [37]. Some years later, the same research group utilized a high-throughput strategy and compared blastocysts produced from women in their forties (*n* = 5) with those derived from young (mean 26.4 years) oocyte donors (*n* = 5). Forty-two DE-miRNAs were identified, with miR-93 being exclusively expressed in blastocysts from women in their forties and up-regulated in aneuploid blastocysts [39].

Finally, Rosenbluth et al., 2013 [38] designed the most comprehensive study to date to identify the most highly expressed miRNAs in human blastocysts. They compared miRNAs expressed in euploid vs. aneuploid embryos, and in male vs. female embryos. They found that the most highly expressed miRNA in euploid embryos was miR-372 and identified 39 miRNAs that were differentially expressed between euploid (*n* = 9) and aneuploid (*n* = 5) embryos, and 21 miRNAs that were differentially expressed between male (*n* = 4) and female (*n* = 5) euploid embryos [38].

## 4. Discussion

To the extent of our knowledge, we have conducted the most comprehensive analysis to date of descriptive and observational studies describing the miRNA content in reproductive female cells/tissues and embryos and the primary associations between miRNA expression and the risk of infertility and compromised early embryo development. On the basis of the results of the present analysis, these data strongly suggest that the presence and expression levels of miRNAs in the aforementioned cells/tissues are related with female fertility and embryo development. These evidences indicate that measuring the expression of particular miRNAs, particularly in exosomes, microvesicles, or in embryo media, might be useful as biomarkers of infertility and may be prognostic of embryo development potential. Increased understanding of the function of these miRNAs may also provide some clues to improve IVF/ICSI protocols and embryo culture.

### 4.1. Highly Expressed miRNAs in Female Reproductive Cells/Tissues and Embryos

Figure 2 shows the most abundantly expressed miRNAs in ovaries, extracellular vesicles (exosomes and microvesicles) and embryos (note that no descriptive studies were found in oocytes). Among the highly expressed miRNAs, three of them are constantly present in ovaries or extracellular vesicles and embryos suggesting a maternal line inheritance.

The hsa-miR-191-5p deserves special attention. The encoding region of this miRNA is located on chromosome 3 (3p21.31) is highly expressed in ovaries and embryos, and seems to play an important role in cell cycle regulation [43]. Moreover, very recently, Cook et al. described that this miRNA, in conjunction with eight others, may serve as a new circulating biomarker of preterm birth [44], which would indicate that it is essential in the tissues analyzed. On the other hand, hsa-miR-202-3p (situated in chromosome 10, q26.3) is interesting because it has been found to play a role in the first and second trimester placenta and fetal growth in two independent studies [45,46]. However, more efforts with this miRNA are required because both reports are relatively small pilot studies. Finally, another remarkable miRNA is hsa-miR-320a (located on chromosome 8, p21.3). This miRNA is highly expressed in ovaries and embryos as well as in epididymis and spermatozoa, suggesting an essential role in both male and female reproductive processes [47]. Mouse knockdowns of this miRNA caused decreased proportions of MII oocytes that developed into two-cell and blastocyst stage embryos [48]. Other authors suggested that miR-320a contributed to ovarian development by targeting *KITLG* [49].

### 4.2. miRNA Deregulations as a Risk of Infertility or Embryo Development Errors

Taking into account the top miRNAs identified as DE-miRNAs (Appendix A) between fertile and infertile women or different embryo types, we propose a list of DE-miRNAs (hsa-miR-9, -21, -27b, -29b, -30a, -146a, -150, -339-3p, -424, -451, -663b, and -1275; Table 3) that can play a central role in female reproductive physiology and embryogenesis.

Although the majority of the aforementioned miRNAs (in Table 3) have a clear association with different cancers and their progression (https://pathcards.genecards.org/card/micrornas_in_cancer), some of these miRNAs also have other functions beyond female fertility. For example, miR-9 has a potential target gene called *TIMP3* that is involved in testis development and differentiation [50]. Another interesting miRNA is miRNA-30a. The miR-30 family, which includes miR-30a, -30b and -30c, has been extensively studied in *Danio rerio* and is associated with the hedgehog pathway, closely related to embryo development [51]. Finally, the inhibition of miR-424 in mice (miR-322 in mice) confirmed the key role of this miRNA in sperm DNA damage during spermatogenesis [52].

### 4.3. Looking Beyond the Present Systematic Review

In the present systematic review, studies dealing with gestational or specific female sexual tract tissues (e.g., fallopian tubes or endometrial tissue) were excluded in order to focus on the expression of miRNAs in ovaries and/or extracellular vesicles, oocytes, and embryos. We encourage the scientific community consider the aforementioned tissues to compliment the conclusions of this systematic review. For example, an interesting study by Ferlita et al. discussed the function of non-coding RNAs (including miRNAs) in endometrial physiology, analyzing their role in endometrial pathologies, such as endometrial cancer, endometriosis, and chronic endometritis [53]. Moreover, mounting evidence suggests that epigenetic changes may play a vital role in both placental-induced diseases, such as pre-eclampsia, and intrauterine growth restriction [54,55].

### 4.4. Concerns, Limitations and Future Directions

The results of this systematic review highlight several important avenues for future research. First, no descriptive studies in oocytes were found in the published literature. This is understandable considering the nature of the biologic material (e.g., bioethical impediments, invasive surgical procedures, etc.); however, more efforts are needed to establish the ‘normal’ miRNA background in this cell type. Second, as we suggested in a previous systematic review of miRNAs on the male reproductive system [47], we emphasize the importance of consistent nomenclature by the use of unequivocal codes such as miRBase Accession Numbers to describe mature miRNAs or the use of the newest codes to distinguish 5p or 3p chains. 

Our study has several limitations. First, given that the systematic review only includes descriptive and observational studies, it is not possible to firmly conclude an existing causal relationship between miRNA expression and the risk of infertility or embryo development. Second, the studies identified are highly heterogeneous in terms of the populations analyzed (e.g., healthy women, PCOS, poor ovarian response, etc.), and for the methodologies used to determine the presence and the expression of miRNAs (e.g., qRT-PCR, microarrays, NGS). Third, the normalization methods applied in the different analysis are again highly heterogeneous. This makes impossible to undertake a meta-analysis of the most representative miRNAs in every tissue, weakening the results. However, the results clearly demonstrate that significant efforts are still required to characterize the role of miRNAs in female reproduction and embryogenesis, and it is our hope that this systematic review will be useful in guiding future research in the field.

Taken together, the 27 included studies provide different evidence levels for the role of miRNAs in female infertility and early embryogenesis. However, it is important to note, that the number of high-quality studies has increased significantly in recent years, as 14 of the 27 articles included in this review (51.8%) were published since 2015. Therefore, this article provides a valuable summary of available data for researchers in the field of reproductive medicine with the aim of highlighting the strengths and weakness of existing data and motivating further progress in this complex field of study.

## 5. Conclusions

The present systematic review of descriptive and observational studies demonstrates the general consensus that aberrant miRNA expression in female reproductive tissues/cells and embryos is related with infertility and embryogenesis errors. Moreover, measuring the expression of particular miRNAs, mainly in extracellular vesicles and embryo media, might become useful biomarkers of infertility and may prove to be prognostic of embryo developmental potential. Further studies with uniform methodologies and large samples are needed to clarify the associations described in the present systematic review.

## Figures and Tables

**Figure 1 cells-08-01564-f001:**
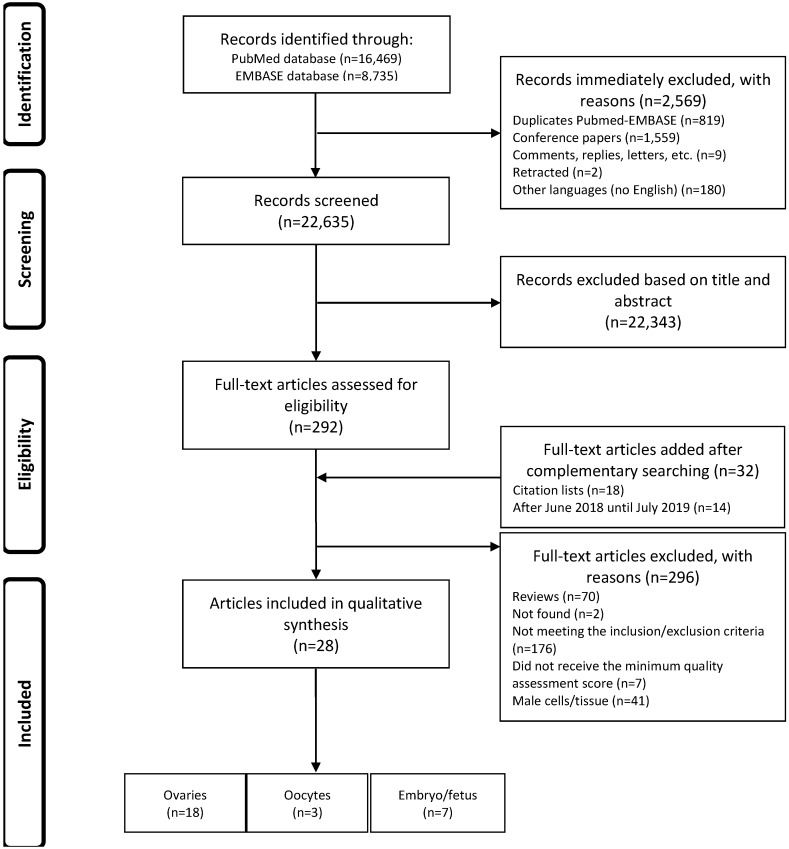
Flow chart of the literature search and selection process.

**Figure 2 cells-08-01564-f002:**
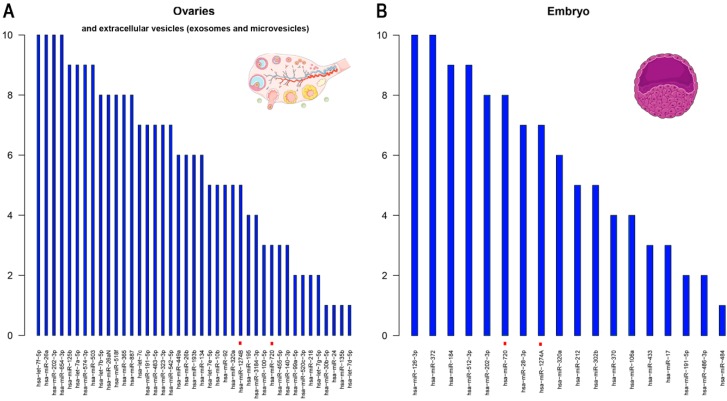
Top expressed miRNAs in human (**A**) ovaries (and/or extracellular vesicles) and (**B**) embryos. Notes: With the information about the top 10 miRNAs detected in every descriptive study we have scored each of the top 10 miRNAs with an arbitrary number (10; the most expressed to 1; the least expressed). The average score was then calculated for each miRNA to create the final list of most highly expressed miRNAs in every tissue/cell type. The miRNAs with potentially equivocal codes were recoded according to miRBase v.22.1 (http://www.mirbase.org/) information. Red dots mean that the miRNA was removed from the miRBase database (not a confirmed miRNA): hsa-miR-720 is a fragment of a tRNA; hsa-miR-1274A is a fragment of fragment of a Lys tRNA; and hsa-miR-1274B is a fragment of fragment of a Lys tRNA.

**Table 1 cells-08-01564-t001:** Summary of the studies in female reproductive cells and tissues: (**A**) ovary and extracellular vesicles (exosomes and microvesicles), and (**B**) oocytes.

Reference	Location	Age (years)	Population Studied	Study Design	Outcome and Aim	Analysis and Normalization Methods	Principal Conclusion	Quality Score
**(A) Ovary and extracellular vesicles (exosomes and microvesicles)**								
[9] (Liang et al., 2007)	USA	ND	1 ovary	Descriptive	To identify miRNA expression patterns in human tissues (including ovary).	Analysis: qRT-PCR. Normalization: Average of miR-30e, miR-92, miR-92N, and miR-423.	A total of 216 miRNAs are expressed in human ovaries.	4/9
[22] (Sang et al., 2013)	China	Desciptive: ND. Cases: 29.09 ± 0.70 Control: 30.83 ± 0.90	Follicular fluid and microvesicles. Descriptive: 20 women undergoing ICSI. Cases: 22 PCOS patients Control: 22.	Descriptive	To identify cell-free miRNAs in human follicular fluid and to investigate the function of these miRNAs in vitro and any roles they play in PCOS.	Analysis: qRT-PCR. Normalization: U6 RNA.	Identification of hsa-miRNA-19b, 24, and 222 in human follicular fluid and another 538 known miRNAs. The expression levels of hsa-miRNA-132 and 320 were significantly lower in controls than in PCOS patients.	4/9
[23] (Santonocito et al., 2014)	Italy	<35	FF exosomes from 15 women undergoing ICSI	Descriptive	To characterize well-represented miRNAs in human FF and exosomes.	Analysis: qRT-PCR. Normalization: Average of miR-25, miR-28-3p, and miR-145.	37 miRNAs upregulated in human FF compared with in plasma (15 miRNAs were found to be upregulated in total FF compared with in plasma; 10 miRNAs were carried by exosomes, while five were not). 22 miRNAs were present exclusively in exosomes.	6/9
[24] (Tong et al., 2014)	China	29.16 ± 2.7	5 women (CRC and COC)	Descriptive	To determine the miRNA expression profiles, via NGS technology, of CRCs and COCs.	Analysis: RNA seq and qRT-PCR validation. Normalization: U6.	A total of 785 and 799 annotated miRNAs were identified in CRCs and COCs respectively. Different expression patterns in CRCs and COCs were detected in 72 annotated miRNAs.	6/9
[16] (Cai et al., 2017)	China	29 ± 3.5	GC. Cases: 25 women with PCOS. Controls 20 women wihtout PCOS.	Case-control	To investigate the effect of miR-145 on cell proliferation and the underlying mechanism of miR-145 in isolated human GCs from the aspirated follicular fluid in women with PCOS.	Analysis: qRT-PCR. Normalization: GAPDH or U6.	miR-145 is downregulated in human GCs from PCOS when compared with control subjects. Insulin receptor substrate 1 (IRS1) gene is a direct target of miR-145.	6/12
[25] (Chen et al., 2017)	China	NOR: 30.57 ± 2.69. DOR: 33.74 ± 3.25	COC. Cases: 10 women with DOR. Controls: 10 women with NOR	Case-control	To comprehensively characterize miRNA expression profiles in cumulus cells of DOR patients.	Analysis: RNA seq and qRT-PCR validation. Normalization: U6.	79 annotated miRNAs, and 5 novel miRNAs were identified differentially expressed between DOR and NOR. mTOR pathway and meiosis-associated biological processes were enriched in the DE-miRNAs	5/12
[27] (Eisenberg et al., 2017)	Israel	<35. PCOS: 26.9 ± 4.3 MIF: 26.8 ± 4.7	GC. Cases: 15 normally ovulating with pure male infertility factor, and 18 with PCOS. Controls: 7 normally ovulating.	Case-control	To study the role of micro-RNA (miRNA)-200b and miRNA-429 in human ovulation and to measure their expression levels in ovulatory and anovulatory patients.	Analysis: qRT-PCR. Normalization: U6.	Very low expression levels were detected for the two miRNAs in granulosa cells, with hsa-miRNA-200b expression relatively higher than miRNA-429. No significant expression difference of any of these miRNAs was identified in the GLCs of the two groups analyzed.	5/12
[28] (Fu et al., 2018)	China	<40. Control: 29.89 ± 3.16. Cases: 30.94 ± 3.82	91 individual FF. Cases: 53 no blastocysts. Controls: 38 viable blastocysts	Case-control	To screen miRNAs in human follicular fluid and to explore the relationship between miRNA expression and blastocyst formation	Analysis: qRT-PCR screening and qRT-PCR validation. Normalization: U6.	13 miRNAs (hsa-miR-185, 202, 22, 224, 29c, 30a-3p, 378, 382, 424, 432, 497, 520c-3p and 663B) were up-regulated, and three miRNAs (hsa-miR-1274a, 139-5p and 150) were down-regulated in cases compared with controls. In the validation, the expression level of miR-663B was found to be significantly higher in cases than in controls.	5/12
[29] (Karakaya et al., 2015)	Turkey	4 groups: <35, 35–37, 38–40, >40.	COC in 189 women undergoing IVF–ICSI. Cases: 24 poor-responders. Controls: 32 non-poor responders.	Case-control	To analyze the association of miRNA expression with the number of oocytes retrieved, in women undergoing in vitro fertilization (IVF).	Analysis: Microarray screening and qRT-PCR validation. Normalization: RNU43.	MicroRNA microarray analysis showed up-regulation of 16 miRNAs and down-regulation of 88 miRNAs in poor responders. qRT-PCR confirmed that miR-21-5p expression was significantly up-regulated in poor responders, whereas miR-21-3p expression was significantly lower.	5/12
[30] (Luo et al., 2019)	China	Poor ovarian response (37 ± 3.16), PCOS (27 ± 3.26), normal patients (29 ± 3.22).	GC. Cases: poor ovarian response (*n* = 7), PCOS (*n* = 20). Controls: normal patients (*n* = 18).	Case-control	To determine the microRNA (miRNA) profiles in GCs from the FF of patients with varying levels of ovarian reserve function.	Analysis: RNA seq.	Identified 20 conserved and 3 novel miRNAs that were upregulated in the poor ovarian response group and 30 conserved miRNAs and 1 novel miRNA that were upregulated in the PCOS group.	7/12
[17] (Roth et al., 2014)	USA	PCOS: 33.1 ± 4.4. Oocyte donors: 27.1 ± 3.6	FF. Cases: PCOS (*n* = 10). Controls: Oocyte donors (*n* = 10).	Case-control	To determine if miRNAs are differentially expressed in the FF of women with PCOS compared to fertile oocyte donors	Analysis: qRT-PCR screening and qRT-PCR validation. Normalization: U6snRNA.	29 miRNAs are differentially expressed between PCOS and OD samples. In the validation step only five of these upregulated miRNAs (hsa-miR-9, 18b, 32, 34c, and 135a) displayed a significant increase in expression in the PCOS group compared to OD controls.	7/12
[18] (Scalici et al., 2016)	France	19–43	FF. Cases: 30 women with PCOS. Controls: 91 women with normal ovarian reserve.	Case-control	To investigate the expression profiles of five circulating miRNAs (let-7b, miR-29a, miR-30a, miR-140 and miR-320a) in human FF	Analysis: qRT-PCR. Normalization: miR-16.	Hsa-miR-30a was significantly up-regulated, while miR-140 and let-7b were significantly down-regulated in FF pools from patients with PCOS (*n* = 30) compared to women with normal ovarian reserve.	6/12
[19] (Shi et al., 2015)	China	Non-PCOS (28.5 ±3.6). PCOS (28.3 ± 3.3).	COC. Cases: PCOS (*n* = 24). Controls: Non-PCOS (*n* = 24).	Case-control	To compare the expression of miRNAs in COC from PCOS and non-PCOS women.	Analysis: qRT-PCR screening and qRT-PCR validation. Normalization: ND.	Hsa-miR-483-5p and 486-5p are significantly decreased in COC of PCOS patients compared with non-PCOS. Four predicted genes, SOCS3, SRF, PTEN and FOXO1, were significantly increased in PCOS COC, and IGF2 was significantly decreased in PCOS COC.	7/12
[20] (Wang et al., 2018)	China	Non-PCOS (30.00 ± 0.74). PCOS (28.67 ± 0.675)	GC. Cases: 25 PCOS. Controls: 17 non-PCOS healthy women.	Case-control	To characterize the function of microRNA-27a-3p (miR-27a-3p) in PCOS	Analysis: qRT-PCR. Normalization: U6 RNA.	21 miRNAs were upregulated and 38 were downregulated in PCOS GCs. Hsa-miR-27a-3p was significantly increased in both excised GCs and the ovaries of patients with PCOS compared with the controls.	6/12
[21] (Xu et al., 2015)	China	Non-PCOS (29.43 ± 3.92). PCOS (28.76 ± 3.51).	GC. Cases: 21 women with PCOS. Controls: 20 women without PCOS.	Case-control	To describe the altered miRNA expression profiles and miRNA targeted signaling pathways in PCOS.	Analysis: RNAseq screening and qRT-PCR validation. Normalization: U6.	A total of 59 known miRNA were identified that are differentially expressed in PCOS cumulus granulosa cells, including 21 miRNAs increased and 38 miRNAs decreased. Notch signaling, regulation of hormone, and energy metabolism of the DE-miRNA target genes.	6/12
[26] (Diez-Fraile et al., 2014)	Belgium	Younger group (29.3–30.3), older group (38.7–42.4)	16 women. Young age: < 31 years (*n* = 8); advanced age: >38 years (*n* = 8)	Prospective	To report the presence of miRNAs in human FF and identify a set of miRNAs that are differentially expressed in older women compared to younger women.	Analysis: qRT-PCR screening and qRT-PCR validation. Normalization: hsa-miR-483-5p.	Hsa-miR-21-5p was present at significantly higher levels in FF from young women, whereas hsa-miR-99b-3p, 134 and 190b were present at significantly higher levels in the FF from older women.	8/14
[32] (Moreno et al., 2015)	Spain	Younger group (32.93 ± 2.19), older group (38.20 ± 0.86)	Stage 1: Younger group: <35 (*n* = 15), older group: >37 (*n* = 15). Stage 2: MII vs. GV (*n* = 12) and MI vs. MII (*n* = 9), each patient acted as her own control.	Prospective	To determine whether there is any difference in the FF miRNA profiles from IVF patients according to their age and oocyte maturation stage.	Analysis: qRT-PCR screening and qRT-PCR validation. Normalization: RNU6B.	hsa-miR-424, which is present in higher proportions in FF from patients with advanced age. When we compared the FF from MII versus GV oocytes, they found 13 differentially expressed miRNAs. When we compared FF from MII versus MI, they found seven differentially expressed miRNAs in MII.	7/14
[31] (Martinez et al., 2018)	USA	28.9–33.7	126 women. EV-miRNAs in FF from oocytes with normal fertilization (*n* = 93) and from oocytes that failed to fertilize (*n* = 33).	Cross-sectional	To assess whether EV miRNAs from FF can serve as biomarkers for fertilization status and day 3 embryo quality.	Analysis: qRT-PCR. Normalization: global mean.	12 EV-miRNAs were differentially expressed between the normal and failed to fertilize groups. Hsa-miR-92a and miR-130b, were over-expressed in FF samples from oocytes that failed to fertilize compared to those that were normally fertilized. Hsa-miR-888 was over-expressed and miR-214 and miR-454 were underexpressed in samples that resulted in impaired day-3 embryo quality compared to top-quality day-3 embryos.	8/14
**(B) Oocytes**								
[33] (Barragán et al., 2017)	Spain	1. Young women with high AFC (age 21 ± 1 years and 24 ± 3 follicles) and low AFC (age 24 ± 2 years and 8 ± 2 follicles); 2. Old women with high AFC (age 32 ± 2 years and 29 ± 7 follicles) and low AFC (age 34 ± 1 years and 7 ± 1 follicles).	36 in vivo matured MII oocytes from 30 healthy women recruited for oocyte donation	Retrospective	To identify the coding and noncoding transcriptional profiles of in vivo matured MII human oocytes and evaluate their changes in relation to age and, independently, ovarian reserve.	Analysis: Microarray and qRT-PCR validation. Normalization: Actin B, ubiquitin C and DNA methyltransferase-1.	A set of five miRNAs (ENSG00000221162, hsa-miR-220b, ENSG00000239174, miR-4262 and 1260a) were increased in old group.	7/14
[35] (Battaglia et al., 2016)	Italy	Young women (28–35 years) and old women (38–40 years)	Six MII oocytes from young women and six from older women	Retrospective	To identify human oocyte miRNAs and demonstrate thatconditions altering oocyte quality, such as reproductive aging	Analysis: qRT-PCR screening and qRT-PCR validation. Normalization: RNU6B.	Twelve miRNAs are differentially expressed in women of advanced reproductive age	7/14
[34] (Xu et al., 2011)	China	1. 30.65 ± 3.87 GV, and 2. 32.15 ± 4.63 MII oocytes	392 oocytes at GV stage and 43 oocytes at MI stage during 251 ICSI cycles	Retrospective	To identify differentially expressed miRNAs and expression patterns of specific miRNAs during meiosis in human oocytes.	Analysis: Microarray and qRT-PCR validation. Normalization: U6 RNA	Compared with GV oocytes, MII oocytes exhibited up-regulation of 4 miRNAs (hsa-miR-193a-5p, 297, 625 and 602), and down-regulation of 11 miRNAs (hsa-miR-888*, 212, 662, 299-5p, 339-5p, 20a, 486-5p, 141*, 768-5p, 376a and 15a).	7/14

Abbreviations: AFC, antral follicle count; COC, cumulus-oocyte complex; CRC, corona radiata cells; DE-miRNAs, differentially expressed miRNAs; DOR, diminished ovarian reserve; EV, extracellular vesicles; FF, follicular fluid; GC, granulosa cells; GLC, granulosa lutein cells; GV, germinal vesicle; ICSI, intracytoplasmic sperm injection; IVF, in-vitro fertilization; MI, metaphase I; MII, metaphase II; ND, no data; NGS, next generation sequencing; NOR, normal ovarian reserve; OD, oocyte donors; PCOS, polycystic ovary syndrome; qRT-PCR, quantitative real-time PCR; RNAseq, RNA sequencing.

**Table 2 cells-08-01564-t002:** Summary of the studies in embryonic tissues.

Reference	Location	Age (years)	Population Studied	Study Design	Outcome and Aim	Analysis and Normalization Methods	Principal Conclusion	Quality Score
[41] (Capalbo et al., 2016)	Italy	ND	ICM-free TE samples and their relative SBM from 5 good-quality human blastocysts	Prospective	To identify miRNAs secreted by human embryos in culture media, which can be used as biomarkers of embryo quality during IVF cycles.	Analysis: qRT-PCR screening and qRT-PCR validation. Normalization: RNU44 and RNU48.	The comparative analysis of TE and SBM samples revealed that 96.6% (57/59) of the miRNAs detected in the SBM were derived from TE cells. MiRNAs analysis of SBM from euploid implanted and unimplanted blastocysts highlighted two miRNAs (hsa-miR-20a and 30c) that showed increased concentrations in the former.	7/14
[42] (Rosenbluth et al., 2014)	USA	ND	28 tested blastocysts SBM, from 13 couples	Prospective	To determine whether human blastocysts secrete miRNAs into culture media and whether these reflect embryonic ploidy status and can predict IVF outcomes.	Analysis: qRT-PCR screening and qRT-PCR validation. Normalization: U6.	Ten miRNAs (hsa-miR-106b, 191, 30c, 372, 376a, 548a-3p, 548c-3p, 548d-3p, 576d-3p, and 603) were consistently detected in the spent IVF culture media, but only two miRNAs (hsa-miR-372 and 191) were confirmed by later single assay qRT-PCR. Hsa-miR-191 was more highly concentrated in media from aneuploid embryos, and hsa-miR-191, 372, and 645 were more highly concentrated in media from failed IVF/non-intracytoplasmic sperm injection cycles.	7/14
[36] (Dominguez et al., 2014)	Spain	Screening: EP: 30.75 ± 1.78, VTOP: 26 ± 3.17. Validation: EP: 30.81 ± 1.55, VTOP: 24.62 ± 1.86.	Cases: 23 patients suffering from tubal EP (8 in screening and 15 in validation). Controls: 29 patients with VTOP (8 in screening and 21 in validation).	Case-control	To investigate the miRNA profile of embryonic tissues in EP and controlled abortions VTOP	Analysis: Microarray screening and qRT-PCR validation. Normalization: SNORD96A.	Four miRNAs (hsa-miR-196b, 30a, 873, and 337-3p) were found to be downregulated in EP versus healthy pregnancy tissues, and three miRNAs (hsa-miR-1288, 451, and 223) were upregulated in EP compared to control pregnancy tissue samples. Validation confirmed the differentially expression of the miRNAs hsa-miR-196 and 223.	7/12
[40] (Lozoya et al., 2014)	Spain	EP: 30.9, VTOP: 21.1	Cases: 17 patients suffering from tubal EP. Controls: 23 patients with VTOP.	Case-control	To determine the expression of the elements of the Lin28/Let-7 system, and related miRNAs in early stages of human placentation and ectopic pregnancy	Analysis: qRT-PCR. Normalization: RNU6.	LIN28B mRNA was barely detectable in embryonic tissue from early stages of gestation and sharply increased thereafter to plateau between gestational weeks 7–9. In contrast, expression levels of Let-7, mir-132 and mir-145 were high in embryonic tissue from early gestations (≤6-weeks) and abruptly declined thereafter, especially for Let-7. Opposite trends were detected for mir-323-3p. Embryonic expression of LIN28B mRNA was higher in early stages (≤6-weeks) of ectopic pregnancy than in normal gestation. In contrast, Let-7a expression was significantly lower in early ectopic pregnancies, while miR-132 and miR-145 levels were not altered. Expression of mir-323-3p was also suppressed in ectopic embryonic tissue.	6/12
[37] (McCallie et al., 2010)	USA	ND	Cryopreserved blastocysts (*n* = 22). Cases: blastocysts from MF infertility alone (*n* = 6), and blastocysts from PCOS women (*n* = 6). Controls: Oocyte donor cycles with no known MF infertility (*n* = 10)	Case-control	To examine human blastocyst miRNA expression (11 probes) in correlation with human infertility.	Analysis: qRT-PCR. Normalization: RNU48.	Morphologically similar blastocysts derived from patients with PCOS or MF infertility exhibited a significant decrease in the expression of six miRNAs (hsa-let-7a, miR-19a, 19b, 24, 93 and 94) in comparison with donor fertile control blastocysts. Annotation of predicted gene targets for these DE-miRNAs included gene ontology biological processes involved in cell growth and maintenance and transcription.	6/12
[39] (McCallie et al., 2014)	USA	Chromosomally normal blastocystsfrom young, OD (26.4 years), Chromosomallynormal blastocysts from women in their forties (40–44 years)	Cases: blastocysts produced from women in their forties (*n* = 5). Controls: young oocyte donor derived blastocysts (*n* = 5)	Case-control	To determine miRNA expression in human blastocysts relative to advanced maternal age and chromosome constitution.	Analysis: qRT-PCR. Normalization: MammU6.	42 DE-miRNAs. miR-93 was exclusively expressed in blastocysts from women in their forties and further up-regulated with an abnormal chromosome complement. Up-regulated miR-93 resulted in an inverse down-regulation of targets like SIRT1, resulting in reduced oxidative defense.	5/12
[38] (Rosenbluth et al., 2013)	USA	ND	Screening (Cases: 5 aneuploids. Controls: 4 male, 5 females euploid.). Validation (Cases: 9 aneuploids. Controls: 7 male, 11 females euploid.).	Case-control and Descriptive	To determine the most highly expressed miRNAs in human blastocysts and to compare miRNAs in euploid versus aneuploid embryos and in male versus female embryos.	Analysis: qRT-PCR screening and qRT-PCR validation. Normalization: snRNA U6.	The most highly expressed miRNA in euploid embryos was miR-372. Screening identified 39 miRNAs that were differentially expressed between euploid (*n* = 9) and aneuploid (*n* = 5) embryos, and 21 miRNAs that were differentially expressed between male (*n* = 4) and female (*n* = 5).	Case-control: 5/12. Descriptive: 6/9.

Abbreviations: DE-miRNAs, differentially expressed miRNAs; EP, ectopic pregnancies; ICM, inner cell mass; IVF, in vitro fertilization; MF, male factor; ND, no data; OD, oocyte donors; PCOS, polycystic ovary syndrome; qRT-PCR, quantitative real-time PCR; SBM, spent blastocyst culture media; TE, trophectoderm; VTOP, voluntary termination of pregnancy.

**Table 3 cells-08-01564-t003:** Primary differentially expressed miRNAs that can play a central role in female reproductive physiology and embryology.

miRNA Name	miRBase Accession Number	Tissue/Cell Expressed	DE-miRNA Condition(s)	Main Publication(s)
hsa-miR-9	MIMAT0000441	1. Granulosa cells 2. Follicular fluid	1. PCOS vs. oocyte donors 2. PCOS vs. oocyte donors	[17,30]
hsa-miR-21	MIMAT0000076	1. Granulosa cells 2. Cumulus cells 3. Follicular fluid	1. PCOS vs. oocyte donors 2. Poor-responders vs. non-poor responders 3. Older vs. younger women	[26,29,30]
hsa-miR-27b	MIMAT0000419	1. Follicular fluid 2. Cumulus cells 3. Blastocysts	1. MII vs. GV 2. Poor-responders vs. non-poor responders 3. Aneuploid vs. euploid	[29,32,38]
hsa-miR-29b	MIMAT0000100	Follicular fluid	MII vs. GV and MII vs. MI	[32]
hsa-miR-30a	MIMAT0000087	1. Follicular fluid 2. Embryonic tissue	1. PCOS vs. oocyte donors 2. EP vs. VTOP	[18,36]
hsa-miR-146a	MIMAT0000449	Granulosa cells	PCOS vs. oocyte donors and Poor ovarian response vs. oocyte donors	[30]
hsa-miR-150	MIMAT0000451	1. Follicular fluid 2. Cumulus cells	1. Non-viable vs. viable blastocysts 2. Poor-responders vs. non-poor responders	[28,29]
hsa-miR-339-3p	MIMAT0004702	1. Follicular fluid 2. Blastocysts	1. MII vs. GV 2. Aneuploid vs. euploid	[32,38]
hsa-miR-424	MIMAT0001341	1. Follicular fluid 2. Cumulus cells 3. Follicular fluid	1. Non-viable vs. viable blastocysts 2. Poor-responders vs. non-poor responders 3. Older vs. younger women	[28,29,32]
hsa-miR-451	MIMAT0001631	1. Follicular fluid 2. Embryonic tissue	1. MII vs. GV and MII vs. MI 2. EP vs. VTOP	[32,36]
hsa-miR-663b	MIMAT0005867	1. Follicular fluid 2. Cumulus cells	1. Non-viable vs. viable blastocysts 2. Poor-responders vs. non-poor responders	[28,29]
hsa-miR-1275	MIMAT0005929	1. Cumulus cells 2. Granulosa cells	1. DOR vs. NOR 2. PCOS vs. oocyte donors	[21,25]

Abbreviations: DOR, diminished ovarian reserve; EP, ectopic pregnancies; GV, germinal vesicle; MI, metaphase I; MII, metaphase II; NOR, normal ovarian reserve; PCOS, polycystic ovary syndrome; VTOP, voluntary termination of pregnancy.

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
