# Peer review of "The Expression of miRNAs in Human Ovaries, Oocytes, Extracellular Vesicles, and Early Embryos: A Systematic Review"

_cells, 2019, doi:10.3390/cells8121564_

Round 1

Reviewer 1 Report

This Review is about the association of aberrant miRNA expression in ovaries, EVs (from the ovaries?), oocytes, and embryos, and fertility/infertility in women. Despite some flaws in the Introduction, the manuscript is clearly structured and well written.

The Introduction does not really lead to the specific topic of the Review.

With respect to the scope of the study, the title should be more specific.

The authors are showing as a main outcome a list of miRNAs likely to be involved in infertility in women. Maybe, it would be interesting to write something about the known roles of those miRNAs and if they are specific for the reproductive system or have rather general functions. I am sure, the reader would be very interested in this information.

Line 54: The authors should cite more review articles about miRNAs and some more recent. The size range is for sure not only from 22-24 nt. Most papers say 18-25, some even between 17 and 27 nt. Looking at human miRNAs in miRbase, the range is between 16 and 28 nt.

Line 57: What is a “semi-complementary configuration”?

Line 59: This should be revised. Cleavage occurs actually only in the case of complete complementary to the target which is the case for siRNAs. The general mechanism is probably rather 2) and 3). For 3), there are many papers showing that this works by interaction/inhibition with/of translation initiation factors.

Line 62: miRbase v22.1 contains 2,654 human matur miRNA sequences.

Line 63: This manuscript doesn’t mention any microRNAs. A later paper by these authors describes RNAs that were probably miRNAs: Ostermeier et al. A suite of novel human spermatozoal RNAs. J Androl. 2005 Jan-Feb;26(1):70-4. We recently described a complex population of spermatozoal coding RNAs that are delivered to the oocyte on fertilization. These are derived throughout spermatogenesis, representing a record of past events. Recently, evidence has been provided that micro-RNAs are present in testes, suggesting that they might also be carried in ejaculate spermatozoa. To directly test this hypothesis, a unique microarray system capable of directly identifying antisense RNAs and predicted transcripts was utilized. RNA isolated from the ejaculate spermatozoa of 6 normal fertile men was directly hybridized to sense oligonucleotide arrays containing 10,000 elements. This revealed 68 shared RNAs, some of which are similar to those previously defined as micro-RNAs, whereas others were the antisense of previously in silico-predicted transcripts. The results and implications of this study are described in this communication.

Line 66ff: There are probably hundreds or thousands of studies exactly showing this. There is no maybe or suggesting etc.

Author Response

November 2019

Dr. Alexander E. Kalyuzhny

Chief Editor of Cells

Dear Dr. Kalyuzhny,

Enclosed please find the revised manuscript #cells-645546 entitled “The expression of miRNAs in human ovaries, oocytes, extracellular vesicles, and early embryos: A systematic review” which we would like to be reconsidered for publication in Cells.

A list of detailed answers to the editor and the reviewers has been included below. We thank all of them for their helpful and thoughtful critique of our manuscript.

We sincerely thank the Editors and Reviewers for the overall appreciation of the submitted manuscript, as well as for the opportunity for our manuscript to be reviewed in Cells and appreciate the valuable and constructive comments on the first version of the manuscript. We have tried to address in detail and accordingly all of the concerns and questions from the reviewers. We provide point-by-point responses to all comments by each of the Reviewers and included the changes (“Track Changes”) and comments in the manuscript, where appropriate.

--------------------------------------------------------------------------------------------------------------------

Reviewer #1:

This Review is about the association of aberrant miRNA expression in ovaries, EVs (from the ovaries?), oocytes, and embryos, and fertility/infertility in women. Despite some flaws in the Introduction, the manuscript is clearly structured and well written.

We sincerely thank Reviewer #1 for the global appreciation of our investigation, as well as for all the valuable comments and suggestions provided in the following lines, which have greatly improved the first version of the manuscript. We have addressed all of them in each of the following points, as well as in the manuscript, when required. Please find below the itemized responses to all of Reviewer #1’s comments.

The Introduction does not really lead to the specific topic of the Review.

As suggested, we have revised the Introduction section. Lines 55-85.

With respect to the scope of the study, the title should be more specific.

We agree with this comment. We have modified the title of the new version of this Manuscript. New title: “The expression of miRNAs in human ovaries, oocytes, extracellular vesicles, and early embryos: A systematic review”.

The authors are showing as a main outcome a list of miRNAs likely to be involved in infertility in women. Maybe, it would be interesting to write something about the known roles of those miRNAs and if they are specific for the reproductive system or have rather general functions. I am sure, the reader would be very interested in this information.

As suggested, we have included these details in Discussion section. Lines 407-416.

Line 54: The authors should cite more review articles about miRNAs and some more recent. The size range is for sure not only from 22-24 nt. Most papers say 18-25, some even between 17 and 27 nt. Looking at human miRNAs in miRbase, the range is between 16 and 28 nt.

As suggested by Reviewer #1, we have adopted the most recent miRBase range: 16-28 nt. Lines 55-56. As suggested, we included 2 new articles about miRNAs.

Line 57: What is a “semi-complementary configuration”?

In animals, mRNA–miRNA interaction is semi-complementary, whereas in plants miRNAs bind with near perfect complementarity on mRNA coding regions. To clarify this, we have modified the sentence. Lines 57-59.

Line 59: This should be revised. Cleavage occurs actually only in the case of complete complementary to the target which is the case for siRNAs. The general mechanism is probably rather 2) and 3). For 3), there are many papers showing that this works by interaction/inhibition with/of translation initiation factors.

Thank you for the correction. We completely agree with this comment. We have clarified this issue in Introduction section. Lines 62-65.

Line 62: miRbase v22.1 contains 2,654 human mature miRNA sequences.

As suggested by Reviewer #1, we have included a new sentence in the MS: “The most recent version of miRBase (v22.1) contains 2,654 mature human miRNA sequences”. Lines 68-69.

Line 63: This manuscript doesn’t mention any microRNAs. A later paper by these authors describes RNAs that were probably miRNAs: Ostermeier et al. A suite of novel human spermatozoal RNAs. J Androl. 2005 Jan-Feb;26(1):70-4. We recently described a complex population of spermatozoal coding RNAs that are delivered to the oocyte on fertilization. These are derived throughout spermatogenesis, representing a record of past events. Recently, evidence has been provided that micro-RNAs are present in testes, suggesting that they might also be carried in ejaculate spermatozoa. To directly test this hypothesis, a unique microarray system capable of directly identifying antisense RNAs and predicted transcripts was utilized. RNA isolated from the ejaculate spermatozoa of 6 normal fertile men was directly hybridized to sense oligonucleotide arrays containing 10,000 elements. This revealed 68 shared RNAs, some of which are similar to those previously defined as micro-RNAs, whereas others were the antisense of previously in silico-predicted transcripts. The results and implications of this study are described in this communication.

As suggested, we have modified the sentence in the MS to include the reference mentioned. Lines 69-71.

Line 66ff: There are probably hundreds or thousands of studies exactly showing this. There is no maybe or suggesting etc.

Changed as suggested. Lines 72-73.

--------------------------------------------------------------------------------------------------------------------

We do hope that all the modifications we have made to the manuscript will make this paper suitable for publication in Cells.

We are looking forward to receiving the editorial decision concerning the newly submitted article.

Yours sincerely,

Dr. Albert Salas-Huetos, and Dr. Marc Yeste

Dr. Albert Salas-Huetos

Andrology and IVF Laboratory, Division of Urology, Department of Surgery, University of Utah School of Medicine, 84180 Salt Lake City, UT, USA. Tel: (+1) 385 210 5534, E-mail: [email protected]

Dr. Marc Yeste

Unit of Cell Biology, Department of Biology, Faculty of Sciences, University of Girona, 17003 Girona, Spain. Tel: (+34) 972 41 95 14, E-mail: [email protected]

Reviewer 2 Report

Title

This is a bit misleading, since the review focuses on mIR expression in selected cell types/tissues within the female reproductive tract, not the entire system. Furthermore, the review provides no insight into function roles suggesting that the words ‘mIRs in female reproduction and embryogenesis’ are not strictly correct. Something more like ‘mIR expression in female reproductive tissues’ may be better.

Abstract

Line 29: delete the word ‘every’ Line 29: Does mIR expression in all reproductive tissues imply an important functional role, or simply indicate a possible functional role? Abstract: I don’t think the rationale is pitched quite right. Is the data ‘highly controversial’, or simply inconclusive?

Introduction

The background and rationale are quite well defined here and this section reads well.

Materials and methods

Why was uterine mIR expression excluded from the review? This would significantly enhance the quality of the manuscript. Was male factor infertility exclusion confirmed in cases of women receiving ICSI/IVF? Line 104: Why were studies investigating ‘free circulating mIRs’ excluded? These can play important functional roles. Line 105: why were studies with gestational or specific female sexual tract tissues (e.g. fallopian tubes or endometrial tissue) excluded, although other pathologies such as PCOS were included? Line 114: What does ‘localization of the study’ mean?

Results

Table 1 – justify inclusion of a study with sample size of n=1 [6; Liang et al 2007] Line 168: Change ovaries to ‘ovary’ The figures are useful although axes text is far too small

Discussion

Limitations of the study are not discussed. The authors correlate expression with function too often throughout the review. What are the future directions - Do mIRs have potential beyond use as biomarkers? What are the implications of changes in mIR expression for reproductive function?

Author Response

November 2019

Dr. Alexander E. Kalyuzhny

Chief Editor of Cells

Dear Dr. Kalyuzhny,

Enclosed please find the revised manuscript #cells-645546 entitled “The expression of miRNAs in human ovaries, oocytes, extracellular vesicles, and early embryos: A systematic review” which we would like to be reconsidered for publication in Cells.

A list of detailed answers to the editor and the reviewers has been included below. We thank all of them for their helpful and thoughtful critique of our manuscript.

We sincerely thank the Editors and Reviewers for the overall appreciation of the submitted manuscript, as well as for the opportunity for our manuscript to be reviewed in Cells and appreciate the valuable and constructive comments on the first version of the manuscript. We have tried to address in detail and accordingly all of the concerns and questions from the reviewers. We provide point-by-point responses to all comments by each of the Reviewers and included the changes (“Track Changes”) and comments in the manuscript, where appropriate.

--------------------------------------------------------------------------------------------------------------------

Reviewer #2:

We sincerely thank Reviewer #2 for the global appreciation of our study, as well as for all the valuable comments and suggestions provided in the following lines, which have greatly improved the first version of the manuscript. We have addressed all of them in each of the following points, as well as in the manuscript, when required. Please find below the itemized responses to the comments.

Title

This is a bit misleading, since the review focuses on mIR expression in selected cell types/tissues within the female reproductive tract, not the entire system. Furthermore, the review provides no insight into function roles suggesting that the words ‘mIRs in female reproduction and embryogenesis’ are not strictly correct. Something more like ‘mIR expression in female reproductive tissues’ may be better.

Following the reviewer’s request, we have modified the title of the new version of this MS. New title: “The expression of miRNAs in human ovaries, oocytes, extracellular vesicles, and early embryos: A systematic review”.

Abstract

Line 29: delete the word ‘every’

Deleted as suggested. Line 30.

Line 29: Does mIR expression in all reproductive tissues imply an important functional role, or simply indicate a possible functional role?

This is an important point. We have modified this issue in the Abstract of the revised MS. Line 31.

Abstract: I don’t think the rationale is pitched quite right. Is the data ‘highly controversial’, or simply inconclusive?

As suggested by the reviewer, we have modified the rationale sentence in the abstract section. Line 33.

Introduction

The background and rationale are quite well defined here and this section reads well.

Thanks for this comment. As suggested by Reviewer #1, we have expanded a little more the Introduction section. Lines 55-85.

Materials and methods

Why was uterine mIR expression excluded from the review? This would significantly enhance the quality of the manuscript. Was male factor infertility exclusion confirmed in cases of women receiving ICSI/IVF? Line 104: Why were studies investigating ‘free circulating mIRs’ excluded? These can play important functional roles. Line 105: why were studies with gestational or specific female sexual tract tissues (e.g. fallopian tubes or endometrial tissue) excluded, although other pathologies such as PCOS were included?

The present systematic review stipulates that “free circulating miRNAs studies, studies in gestational or specific female sexual tract tissues (e.g. fallopian tubes or endometrial tissue) were excluded”, in order to focus the review on the expression of miRNAs in ovaries and/or extracellular vesicles, oocytes and embryos. However, in order to suggest to the scientific community other possible systematic reviews that would compliment the conclusions of the present systematic review, we have added a new section in the Discussion stating that “Looking beyond the present systematic review…”. Lines 418-429.

Line 114: What does ‘localization of the study’ mean?

It is interesting to know the localization of the cohort to address whether the results are geographically biased. As the reviewer can see, there are articles throughout all the world with an emphasis on the USA and China.

Results

Table 1 – justify inclusion of a study with sample size of n=1 [6; Liang et al 2007]

We included this reference because it is the first well-designed, descriptive study published. However, we have clarified in the new version of this MS that “the results of this study must be cautiously interpreted due to the limited sample size”. Lines 178-179.

Line 168: Change ovaries to ‘ovary’

Changed as suggested.

The figures are useful although axes text is far too small

As suggested, we enlarge the figure axes text of the Figure 2.

Discussion

Limitations of the study are not discussed. The authors correlate expression with function too often throughout the review. What are the future directions - Do mIRs have potential beyond use as biomarkers? What are the implications of changes in mIR expression for reproductive function?

Thank you for this comment. We respectfully disagree with the reviewer’s assessment. In the subsection entitled “4.4 Concerns, limitations and future directions”, we discuss in detail the limitations of the study, future directions and potential implications. Please see lines 431-459.

--------------------------------------------------------------------------------------------------------------------

We do hope that all the modifications we have made to the manuscript will make this paper suitable for publication in Cells.

We are looking forward to receiving the editorial decision concerning the newly submitted article.

Yours sincerely,

Dr. Albert Salas-Huetos, and Dr. Marc Yeste

Dr. Albert Salas-Huetos

Andrology and IVF Laboratory, Division of Urology, Department of Surgery, University of Utah School of Medicine, 84180 Salt Lake City, UT, USA. Tel: (+1) 385 210 5534, E-mail: [email protected]

Dr. Marc Yeste

Unit of Cell Biology, Department of Biology, Faculty of Sciences, University of Girona, 17003 Girona, Spain. Tel: (+34) 972 41 95 14, E-mail: [email protected]

Reviewer 3 Report

I read with great interest the Manuscript entitled “MiRNAs in female human reproduction and early embryo development: A systematic review” (cells-645546).

It is an interesting review about the role of miRNAs in female human reproduction and early embryogenesis. The review is well written, has an important clinical message, and should be of great interest to the readers of Cells. The Manuscript can be further expanded and improved, and reference list can be updated by citing recent studies about the topic.

According to my opinion, only a few small improvements are needed, as suggested below:

An interesting study by Ferlita et al. (Int J Mol Sci. 2018 Jul 20;19(7). pii: E2120. doi: 10.3390/ijms19072120) discussed the function of non-coding RNAs in endometrial physiology, analysing their role in endometrial pathologies such as endometrial cancer, endometriosis and chronic endometritis. I suggest further discussing this point starting from the above-mentioned study. Recent and novel evidence suggested that epigenetic changes, in particular altered expression of selective miRNA, may play a key role in both placental-induced diseases such pre-eclampsia and intrauterine growth restriction. It would be interesting to discuss (at least briefly) this topic, referring to: Biomed Res Int. 2017;2017:6073167. doi: 10.1155/2017/6073167; Expert Rev Mol Diagn. 2015;15(8):999-1010. doi: 10.1586/14737159.2015.1053468.

Author Response

November 2019

Dr. Alexander E. Kalyuzhny

Chief Editor of Cells

Dear Dr. Kalyuzhny,

Enclosed please find the revised manuscript #cells-645546 entitled “The expression of miRNAs in human ovaries, oocytes, extracellular vesicles, and early embryos: A systematic review” which we would like to be reconsidered for publication in Cells.

A list of detailed answers to the editor and the reviewers has been included below. We thank all of them for their helpful and thoughtful critique of our manuscript.

We sincerely thank the Editors and Reviewers for the overall appreciation of the submitted manuscript, as well as for the opportunity for our manuscript to be reviewed in Cells and appreciate the valuable and constructive comments on the first version of the manuscript. We have tried to address in detail and accordingly all of the concerns and questions from the reviewers. We provide point-by-point responses to all comments by each of the Reviewers and included the changes (“Track Changes”) and comments in the manuscript, where appropriate.

--------------------------------------------------------------------------------------------------------------------

Reviewer #3:

I read with great interest the Manuscript entitled “MiRNAs in female human reproduction and early embryo development: A systematic review” (cells-645546).

It is an interesting review about the role of miRNAs in female human reproduction and early embryogenesis. The review is well written, has an important clinical message, and should be of great interest to the readers of Cells. The Manuscript can be further expanded and improved, and reference list can be updated by citing recent studies about the topic.

We sincerely thank Reviewer #3 for the global appreciation of our study, as well as for all the valuable comments and suggestions provided in the following lines, which have greatly improved the first version of the MS. We have addressed all of them in each of the following points, as well as in the MS, when required. Please find below the itemized responses to the comments.

According to my opinion, only a few small improvements are needed, as suggested below:

An interesting study by Ferlita et al. (Int J Mol Sci. 2018 Jul 20;19(7). pii: E2120. doi: 10.3390/ijms19072120) discussed the function of non-coding RNAs in endometrial physiology, analysing their role in endometrial pathologies such as endometrial cancer, endometriosis and chronic endometritis. I suggest further discussing this point starting from the above-mentioned study.

Recent and novel evidence suggested that epigenetic changes, in particular altered expression of selective miRNA, may play a key role in both placental-induced diseases such pre-eclampsia and intrauterine growth restriction. It would be interesting to discuss (at least briefly) this topic, referring to: Biomed Res Int. 2017;2017:6073167. doi: 10.1155/2017/6073167; Expert Rev Mol Diagn. 2015;15(8):999-1010. doi: 10.1586/14737159.2015.1053468.

We agree with the Reviewer. However, because the present systematic review stipulates that “studies in gestational or specific female sexual tract tissues (e.g. fallopian tubes or endometrial tissue) were excluded”, in order to focus the review on the expression of miRNAs in ovaries and/or extracellular vesicles, oocytes and embryos, we have added a new section in the Discussion part: “Looking beyond the present systematic review”. Lines 418-429.

--------------------------------------------------------------------------------------------------------------------

We do hope that all the modifications we have made to the manuscript will make this paper suitable for publication in Cells.

We are looking forward to receiving the editorial decision concerning the newly submitted article.

Yours sincerely,

Dr. Albert Salas-Huetos, and Dr. Marc Yeste

Dr. Albert Salas-Huetos

Andrology and IVF Laboratory, Division of Urology, Department of Surgery, University of Utah School of Medicine, 84180 Salt Lake City, UT, USA. Tel: (+1) 385 210 5534, E-mail: [email protected]

Dr. Marc Yeste

Unit of Cell Biology, Department of Biology, Faculty of Sciences, University of Girona, 17003 Girona, Spain. Tel: (+34) 972 41 95 14, E-mail: [email protected]

Reviewer 4 Report

In this review, the authors present results of descriptive and observational studies dealing with miRNA expression in human ovaries, oocytes, extracellular vesicle and embryos in relation to reproductive disorders, fertilization, embryo quality and implantation. The authors excluded the vast majority of papers acquired through the primary search in databases on the basis of objective, but also subjective (quality assessment) criteria. The data of the present study suggest that the presence and expression levels of miRNAs in the above mentioned cells/tissues are related with female fertility and embryo development and can thus be used as predictive markers of human infertility. As such, the review provides valuable information on the presence and potential role of the most important miRNAs identified in ovarian tissues, oocytes and early embryos. In general, it is well designed and well-written review in an important and recent topic.

The primary search of databases was obviously conducted with the aim to identify papers dealing with the role of miRNA in both male and female reproduction. The authors used the same systematic search protocol (PROSPERO 2018: CRD42018099793) as in the previous male-related research (Salas-Huetos et al., Andrology 2019, in press). I am not sure if this approach complies with the Cells publishing policies. This approach may also be one of the reasons of the dramatic reduction of more than 22 thousand screened papers to merely 27 analyzed papers, which does not seem to cover this research area exhaustively. On the other hand, my own more focused search matched to the large extent the list of papers analyzed in this review. This suggests that the role of miRNAs in regulation of female human fertility has been insufficiently investigated until recently. The present review may thus encourage more scientists to pursue research in this field.

Specific comments:

1- Page 3 Line 104: Please, define "RCT studies".

2- Page 5 Line 153 vs. Page 13 Line 164: Is it 12 or 18 studies? In Fig1 and Page5 line148, it is mentioned as 18 studies in ovaries and/or extracellular vesicles. Please check and correct.

3- Regarding the oocyte part, the review includes only 2 studies. Why the authors did not include other studies, for example, Battaglia et al. 2016, Biology of Reproduction 95:131 "MicroRNAs are stored in human MII oocyte and their expression profile changes in reproductive aging", which I think fits well with the scope of this review.

Author Response

November 2019

Dr. Alexander E. Kalyuzhny

Chief Editor of Cells

Dear Dr. Kalyuzhny,

Enclosed please find the revised manuscript #cells-645546 entitled “The expression of miRNAs in human ovaries, oocytes, extracellular vesicles, and early embryos: A systematic review” which we would like to be reconsidered for publication in Cells.

A list of detailed answers to the editor and the reviewers has been included below. We thank all of them for their helpful and thoughtful critique of our manuscript.

We sincerely thank the Editors and Reviewers for the overall appreciation of the submitted manuscript, as well as for the opportunity for our manuscript to be reviewed in Cells and appreciate the valuable and constructive comments on the first version of the manuscript. We have tried to address in detail and accordingly all of the concerns and questions from the reviewers. We provide point-by-point responses to all comments by each of the Reviewers and included the changes (“Track Changes”) and comments in the manuscript, where appropriate.

--------------------------------------------------------------------------------------------------------------------

Reviewer #4:

In this review, the authors present results of descriptive and observational studies dealing with miRNA expression in human ovaries, oocytes, extracellular vesicle and embryos in relation to reproductive disorders, fertilization, embryo quality and implantation. The authors excluded the vast majority of papers acquired through the primary search in databases on the basis of objective, but also subjective (quality assessment) criteria. The data of the present study suggest that the presence and expression levels of miRNAs in the above mentioned cells/tissues are related with female fertility and embryo development and can thus be used as predictive markers of human infertility. As such, the review provides valuable information on the presence and potential role of the most important miRNAs identified in ovarian tissues, oocytes and early embryos. In general, it is well designed and well-written review in an important and recent topic.

We sincerely thank Reviewer #4 for the global appreciation of our study, as well as for all the valuable comments and suggestions provided in the following lines, which have greatly improved the first version of this Manuscript. We have addressed all of them in each of the following points, as well as in the MS, when required. Please find below the itemized responses to the comments.

The primary search of databases was obviously conducted with the aim to identify papers dealing with the role of miRNA in both male and female reproduction. The authors used the same systematic search protocol (PROSPERO 2018: CRD42018099793) as in the previous male-related research (Salas-Huetos et al., Andrology 2019, in press). I am not sure if this approach complies with the Cells publishing policies. This approach may also be one of the reasons of the dramatic reduction of more than 22 thousand screened papers to merely 27 analyzed papers, which does not seem to cover this research area exhaustively. On the other hand, my own more focused search matched to the large extent the list of papers analyzed in this review. This suggests that the role of miRNAs in regulation of female human fertility has been insufficiently investigated until recently. The present review may thus encourage more scientists to pursue research in this field.

Thanks for asking us about this issue. Our initial idea was to publish a really comprehensive review of the role of miRNAs in human reproduction; however, in the revision step of a previous article recently published (Salas-Huetos et al., Andrology 2019, in press) two of the reviewers suggested that we divide the article in two: “The extent of the tissues selected was very broad and could be easily divided into two distinct analyses one on male and one on female reproductive tissues. I suggest that focusing on male reproductive tissues will be adequate for a systematic review, where additional studies could be included.

Following these suggestions, we decided to divide the article into two, but maintained the same systematic protocol to be able to compare both articles as one.

Specific comments:

1- Page 3 Line 104: Please, define "RCT studies".

Done as suggested. Line 113.

 2- Page 5 Line 153 vs. Page 13 Line 164: Is it 12 or 18 studies? In Fig1 and Page5 line148, it is mentioned as 18 studies in ovaries and/or extracellular vesicles. Please check and correct.

Thanks. Corrected as suggested. Lines 163 and 174.

3- Regarding the oocyte part, the review includes only 2 studies. Why the authors did not include other studies, for example, Battaglia et al. 2016, Biology of Reproduction 95:131 "MicroRNAs are stored in human MII oocyte and their expression profile changes in reproductive aging", which I think fits well with the scope of this review.

We agree with the Reviewer #4 that this article could be interesting within the scope of the present review. While, according our records, the article should not be included in the first step (identification), we believe, following the reviewer’s comment, that it deserves to be included. Changed as suggested: Figure 1, Table 1, Table S2 and lines 41-42, 144, 146-147, 152, 158-159, 274, 288-292.

--------------------------------------------------------------------------------------------------------------------

We do hope that all the modifications we have made to the manuscript will make this paper suitable for publication in Cells.

We are looking forward to receiving the editorial decision concerning the newly submitted article.

Yours sincerely,

Dr. Albert Salas-Huetos, and Dr. Marc Yeste

Dr. Albert Salas-Huetos

Andrology and IVF Laboratory, Division of Urology, Department of Surgery, University of Utah School of Medicine, 84180 Salt Lake City, UT, USA. Tel: (+1) 385 210 5534, E-mail: [email protected]

Dr. Marc Yeste

Unit of Cell Biology, Department of Biology, Faculty of Sciences, University of Girona, 17003 Girona, Spain. Tel: (+34) 972 41 95 14, E-mail: [email protected]

Round 2

Reviewer 1 Report

No further comments

Reviewer 2 Report

The authors have addressed my concerns in their revised version of this review.